# NaCl-Templated Ultrathin 2D-Yttria Nanosheets Supported Pt Nanoparticles for Enhancing CO Oxidation Reaction

**DOI:** 10.3390/nano12132306

**Published:** 2022-07-05

**Authors:** Luozhen Jiang, Chen Tian, Yunan Li, Rui Si, Meng Du, Xiuhong Li, Lingling Guo, Lina Li

**Affiliations:** Shanghai Advanced Research Institute, Chinese Academy of Sciences, Shanghai 201210, China; lzjiang@mail.ustc.edu.cn (L.J.); tianchen@sinap.ac.cn (C.T.); liyunan@sinap.ac.cn (Y.L.); sirui@mail.sysu.edu.cn (R.S.); dum@sari.ac.cn (M.D.)

**Keywords:** platinum nanoparticles, NaCl templated, 2D-yttria nanosheet, CO oxidation, X-ray absorption fine structure

## Abstract

Morphology of support is of fundamental significance to the fabrication of highly efficient catalysts for CO oxidation reaction. Many methods for the construction of supports with specific morphology and structures greatly rely on controlling general physical and chemical synthesis conditions such as temperature or pH. In this paper, we report a facile route to prepare yttria nanosheet using NaCl as template to support platinum nanoparticles exhibiting higher CO oxidation activity than that of the normally prepared Pt/Y_2_O_3_. With the help of TEM and SEM, we found that Pt NPs evenly distributed on the surface of NaCl modified 2D-nanosheets with smaller size. The combination of XAFS and TEM characterizations demonstrated that the nano-size Pt species with Pt*_x_*O*_y_* structure played an essential role in the conversion of CO and kept steady during the CO oxidation process. Moreover, the Pt nanoparticles supported on the NaCl templated Y_2_O_3_ nanosheets could be more easily reduced and thus exposed more Pt sites to adsorb CO molecules for CO oxidation according to XPS and DRIFTS results. This work offers a unique and general method for the preparation of potential non-cerium oxide rare earth element oxide supported nanocatalysts.

## 1. Introduction

The CO oxidation reaction (CO + 1/2O_2_ = CO_2_) is a typical model reaction in fundamental catalysis research, and it also serves as a significant step in resolving exhaust pollution in the atmosphere [1,2,3,4]. To the best of our knowledge, rational design of catalysts is a key step for catalysis research [5]. Platinum(Pt)-based noble metal catalysts have been widely studied in promoting CO oxidation reaction deriving from their outstanding catalytic activity [6,7,8] and, at the same time, various materials have been explored for the design and fabrication of supported platinum catalysts, especially dominated by irreducible oxide supports such as SiO_2_ [9,10] and Al_2_O_3_ [11,12], and reducible oxides—including CeO_2_ [13,14], FeO*_x_* [15], ZnO [16,17], and so on [18]. For irreducible oxides supported catalysts, it is generally considered that the relationship between the catalytic activity and the intrinsically active sites of platinum species can be exactly evaluated, while it is difficult to establish strong interaction between platinum species and inert support like SiO_2_; thus, the supported platinum species are easily agglomerated during the high-temperature or long-term CO oxidation process, leading to a dramatic decrease in CO oxidation activity [19]. On the other hand, the reducible oxides with variable oxidation states or rich surface oxygen vacancy usually tend to construct high-performance nano-catalysts, and even atomically dispersed platinum by the strong interaction between metal and support [20]. However, rare earth oxides other than cerium oxide are rarely adopted in CO oxidation reaction, and even if they are available, their activity deactivates rapidly as a result of generating carbonate species during the reaction, according to our previously reported lanthanum-supported platinum catalysts [21]. Therefore, exploring other methods to narrow the activity gap between cerium oxide and the non-cerium oxide rare-earth-element-oxide-supported Pt catalysts by modifying or regulating the structure of non-cerium oxide rare earth element oxides is still desired for CO oxidation reaction.

Yttrium oxide (Y_2_O_3_), as one of the most well-known thermodynamically stable rare earth oxides, has been applied in a wide range of catalytic systems acting as supports or promotors—including propylene epoxidation, CO and CO_2_ methanation, N_2_O decomposition, propane dehydrogenation, methane conversion to syngas [22], alkaline methanol electrooxidation, and so on. In the case of CO_2_ methanation reaction, Y_2_O_3_ was directly served as a support [23]. Yan et al. studied the differences of Y-precursors in fabricating Ni/Y_2_O_3_ catalysts with different interactions between Ni and the Y_2_O_3_ matrix, and thus greatly improved the catalytic activity and stability in this reaction with enhanced CO anti-poisoning capability. Li et al. reported greatly increased performance of methanol electrooxidation by constructing a strong electronic effect between the Pd and Y_2_O_3_ to improve the Pd catalyst dispersion [24]. Ryoo et al. even incorporated Y along with Pt into a mesoporous zeolite to form Pt3Y intermetallic nanoparticles by heating under H_2_ flow at 700 °C, and which was found to be more stable, active, and selective than that of monometallic Pt catalysts for propane dehydrogenation reaction [25]. In addition to the interaction between metal and Y_2_O_3_ support, the structural morphology of Y_2_O_3_ also severely affected the catalytic performance of such Y_2_O_3_-containing catalysts. Guzman et al. investigated that the nanocrystalline Y_2_O_3_ could stabilize more active gold species in the process of synthesis Au/Y_2_O_3_ catalyst than mesostructured and precipitated Y_2_O_3_, and as a result of increased CO oxidation activity [26]. Sreethawong et al. further explored the structural transformation of yttrium hydroxide (Y(OH)_3_) to yttrium oxide (Y_2_O_3_) along with the increasing of calcination temperature for CO oxidation and denoted that both the yttrium oxide hydroxide (YOOH) and the yttrium-oxide-supported Au catalyst displayed the highest catalytic activity in this reaction attribute to the oxygen mobilization [27]. However, the application of Y_2_O_3_ as a support for the design of Pt catalysts over CO oxidation reaction has not been reported so far, probably due to the difficulty in rationally stabilizing highly dispersed Pt nanoparticles (NPs) towards Y_2_O_3_ matrix. 

In this work, we firstly report the synthesis of NaCl-templated ultrathin yttrium trioxide nanosheets-immobilized Pt NPs using an initial wet-impregnation method with Pt loading of 0.7 wt %. Herein, cubic NaCl serves as an effective and universal directed agent for synthesizing the regular and ultrathin Y_2_O_3_ nanosheets [28]. It was interesting that the Pt species anchored on NaCl-templated Y_2_O_3_ nanosheets exhibited remarkably higher catalytic activity for CO oxidation reaction than the non-templated counterpart, featuring the total CO conversion from 214 °C to 188 °C. The promoted reactivity can be ascribed to formation of very stable and highly dispersed Pt NPs on the NaCl-oriented Y_2_O_3_ nanosheets.

## 2. Materials and Methods

### 2.1. Catalyst Preparation

Y(NO_3_)_3_·6H_2_O, [Pt(NH_3_)_4_](NO_3_)_2_ and LiF were all purchased from Sinopharm Chemical Reagent Co., Ltd. without any subsequent purification. The gas was obtained from Air Liquide Co., keeping a purity of 99.997%.

NaCl cube template—3.00 g of NaCl was dissolved in 10 mL deionized water, and then dropwise added to 100 mL absolute ethyl alcohol slowly under stirring and stirred continuously for 30 min at room temperature. The white precipitates were filtered and washed with absolute ethyl alcohol dried under 60 °C for 12 h to obtain NaCl cube template.

The Y_2_O_3_ support—0.33 g of Y(NO_3_)_3_·6H_2_O were dissolved in 10 mL absolute ethyl alcohol and 6.00 g of NaCl cube template were added in this solution. The solution was dried by evaporation at 60 °C. Then, the white solid was calcined at 400 °C for 10 h. After cooling to room temperature, the solid was washed with deionized water 3 times to remove NaCl. The remaining solid was dried at 60 °C for 12 h to obtain 2D-Y_2_O_3_ nanosheet supports marked Y_2_O_3_-NaCl. 0.33 g of Y(NO_3_)_3_·6H_2_O was directly calcined at 400 °C for 4 h to obtain Y_2_O_3_ support marked Y_2_O_3_.

The Pt/Y_2_O_3_ catalysts—The catalysts were obtained using an initial wet-impregnation route. A measure of 14 mg of [Pt(NH_3_)_4_](NO_3_)_2_ was dissolved in 10 mL H_2_O and then added to 1.00 g of supports and thereafter ultrasonically vibrated for 60 min. The aqueous was then dried by distillation at 80 °C in air and the precipitation was dried at 80 °C for 12 h. Then, the 0.7 wt % Pt/Y_2_O_3_ and Pt/Y_2_O_3_-NaCl catalyst was obtained after calcination in air at 400 °C for 4 h.

### 2.2. Characterization Methods

The bulk concentrations of Pt of the catalysts were determined by inductively coupled plasma atomic emission spectroscopy (ICP-AES) on an IRIS Intrepid II XSP instrument (Thermo Electron Corporation, Waltham, MA, USA).

The powder X-ray diffraction (XRD) test was carried out on a Bruker D8 Advance diffractometer (40 kV and 40 mA) featuring a Cu *K*_a1_ radiation (*λ* = 1.5406 Å) source and using a scanning rate of 4 (°)/min. The XRD patterns were collected from 10° to 90° with a step of 0.02°. A mm-scale α-Al_2_O_3_ disc was applied to calibrate 2*θ* angles. The fresh or used sample was placed into a quartz sample holder for each test.

The nitrogen adsorption–desorption measurements were performed on an ASAP2020-HD88 analyzer (Micromeritics Co., Ltd., Atlanta, GA, USA) at 77 K. Before analysis, the synthesized powder sample was degassed at 250 °C under vacuum (<100 μm Hg) for 4 h. The BET-specific surface areas (*S*_BET_) were obtained based on the relative pressure (*P*/*P*_0_) range between 0.05 and 0.20. 

The scanning electron microscope (SEM) images were examined via focused ion beam scanning electron microscopy (Zeiss, Cross Beam 540).

X-ray photoelectron spectroscopy (XPS) was performed by PHI 5000 VersaProbe III with a monochromatic Al Kα X-ray source with the beam size of 100 μm × 1400 μm. Charge compensation was achieved via dual beam charge neutralization and the binding energy was corrected by setting the binding energy of the hydrocarbon C 1s feature to 284.8 eV. The curve fitting was performed by PHI MultiPak software, and Gaussian-Lorentz functions and Shirley background were used.

The TEM experiments were performed on a FEI Tecnai G^2^ F20 microscope (FEI Co., Hillsboro, TX, USA) operating at 200 kV. The tested sample was prepared by suspending in ethanol, and then a drop of the dilute suspension was cast on a carbon film-coated Mo grid. The fresh sample grid was dried enough before loaded into the specific TEM sample holder.

The X-ray absorption fine structure (XAFS) spectra at the Pt L3 (E_0_ = 11,564 eV) edge were collected at BL11B1 beamline of the Shanghai Synchrotron Radiation Facility (SSRF) operated at the energy of storage ring of 3.5 GeV in ‘top-up’ mode with a constant current of 200 mA. To obtain quasi in-situ XAFS results, the reduced or used samples were sealed in a stainless reactor with two globe valves and directly transferred to a glove box under nitrogen atmosphere to ensure that the samples were not exposed to air during the tableting process before the XAFS test. The XAFS data were recorded under fluorescence mode using a Lytle detector for Pt-based catalysts. The absorption edge of Pt foil was applied to calibrate the X-ray energy. The collected XAFS data were extracted by Athena and the corresponding profiles were fitted by Artemis codes. For the X-ray absorption near edge structure (XANES) section, the experimental absorption coefficients as a function of energies *μ*(E) were obtained by background subtraction and normalization procedures and finally reported as ‘normalized absorption’. The valences of platinum were acquired by linear combination fitted by using the corresponding references (Pt foil for Pt^0^, PtO_2_ for Pt^4+^). For the extended X-ray absorption fine structure (EXAFS) section, the Fourier transformed (FT) data in R space were calculated by using PtO_2_ and metallic Pt model for Pt-O and Pt-Pt contributions. The passive electron factors *S*_0_^2^ were acquired by fitting the experimental data on Pt foils and fixing Pt-Pt coordination number (*CN*) to be 12, and later fixed for further analysis of the analyzed samples. The descriptive parameters of electronic properties (e.g., correction to the photoelectron energy origin, *E*_0_) and local structure environment including *CN*, bond distance (*R*), and Debye–Waller factor around the absorbing atoms were allowed to vary in the fitting process. The fitted ranges for k and R spaces were selected as k = 3–11 Å^−1^ with R = 1–3 Å (*k*^2^ weighted). 

In situ diffuse reflectance infrared Fourier transform spectroscopy (DRIFTS) was performed on a Bruker Vertex 70 FTIR spectrometer equipped with a mercury-cadmium-telluride detector. Before testing, approximately 30 mg fresh powder samples were spread evenly in the reaction cell, and pretreated at 300 °C under H_2_ atmosphere for 30 min. Then, the samples were cooled down to room temperature (RT), and the reduced gas was switched to pure He (30 mL·min^−1^). The ‘CO-He-CO-O_2_’ steady-state mode was usually carried out as a typical recording process for in situ DRIFTS. In our study, 190 °C were selected as characteristic test temperatures. Taking the 190 °C steady-state test as an example, after He (30 mL·min^−1^) purging for 30 min at RT, the temperature of the sample cell was warmed up to 190 °C and kept steady, and the background spectra were recorded with a resolution of 4 cm^−1^. The real-time recording of spectra was started once the testing sample exposed to 5% CO/He (30 mL·min^−1^). Next, 60 min later, the CO mixture flow was switched to an He stream (30 mL·min^−1^) and purged for 60 min. Then, the sample exposed to 5% CO/He (30 mL·min^−1^) again for 60 min. Finally, the same amount of 5%O_2_/He (30 mL·min^−1^) was introduced with a gas composition of 1%CO/20%O_2_/He (30 mL·min^−1^, 1 atm) for 60 min. The DRIFTS spectra were obtained by subtracting the background from the measured sample data. Moreover, in situ DRIFTS under real reaction condition were also collected—100 °C, 150 °C, and 200 °C were particularly analyzed.

### 2.3. Catalytic Tests

The CO oxidation activities of Pt/Y_2_O_3_ catalysts (30 mg) were investigated in a fixed-bed reactor, and the concentrations of typical CO and CO_2_ in the output mixture were detected online by a Gas analyzer (Cubic-Ruiyi Co., Ltd., Wuhan, China). The feeding gas was a gas mixture of 1%CO/20%O_2_/He (60 mL·min^−1^, 99.999% purity from Air Liquide Co., Ltd., Shanghai, China) with a corresponding space velocity (SV) of 120,000 mL·h^−1^·g_cat_^−1^. Before CO oxidation testing, the samples were reduced by 5% H_2_/N_2_ (60 mL·min^−1^) at 300 °C for almost 30 min and subsequently cooled down to 30 °C in pure He (60 mL·min^−1^). The ‘light-off’ test was ramped from RT to 250 °C with a heating rate of 5 °C·min^−1^ under the set reaction condition. The stability tests of the catalysts were analyzed at 200 °C and 180 °C for 11 h. For kinetic tests, CO conversions were tuned within 20% to exclude the internal and external diffusion limitations.

## 3. Results and Discussion 

### 3.1. Structural Characterization of Fresh Platinum-Yttrium Oxide Catalysts

For the purpose of exploring catalytic activities of yttrium oxide-supported platinum (Pt/Y_2_O_3_) catalysts in CO oxidation and significantly improve the reactivity of such catalysts, sodium chloride (NaCl) was selected as a template to optimize the properties of the support, and an incipient wetness impregnation method was used to prepare the supported catalysts with and without NaCl templated, namely Pt/Y_2_O_3_-NaCl and Pt/Y_2_O_3_ (Figure 1). The Y(NO_3_)_3_·6H_2_O would tend to attach on the surface of NaCl cube after ethanol was evaporated off; and once calcined at 400 °C, the Y(NO_3_)_3_·6H_2_O decomposed to form 2D-Y_2_O_3_. The Pt^2+^ ion tend to evenly disperse on the surface of the 2D-flat and after calcination at 400 °C, the mellitic Pt species prefer to disperse uniformly on the surface of the 2D-flat, while rough or irregular Y_2_O_3_ supports are prone to form mixed phases containing Pt and PtO_2_ species, as evidenced by the following characterization.

Figure 2a is an SEM image of pure NaCl crystals obtained by washing with copious amounts of absolute ethyl alcohol that shows uniform and regular microcubes. According to previous reports that NaCl-assisted templating was a proven method for synthesizing ultrathin 2D oxides [28], our NaCl-templated Y_2_O_3_ support labeled Y_2_O_3_-NaCl was fabricated through a one-pot pyrolysis containing yttrium nitrate and NaCl, and a subsequent template removal process. The average crystallite size measured from XRD patterns of Y_2_O_3_ was increased to 30 nm from 17 nm after the adoption of a NaCl template (Table 1), indicating the growth of Y_2_O_3_ lattice was along the surface of the NaCl to form a more regular shape. For comparison, another Y_2_O_3_ support without a NaCl template was obtained by direct pyrolysis of yttrium nitrate. As shown in Figure 2b, the Pt/Y_2_O_3_-NaCl sample displays obvious layered structures, while the Pt/Y_2_O_3_ sample without a NaCl template shows large particles formed by agglomeration of many irregular shaped structures, indicating that the preparation of Y_2_O_3_ nanosheets can be realized by the NaCl-templated method, and the structural differences in yttrium oxide support may play a crucial role in the dispersion of Pt species.

The N_2_ adsorption isotherms of the two kinds of Y_2_O_3_ support are shown in Figure 3a, the curves are close to IV type isotherms and corresponded to H3 hysteresis loop [22,29], demonstrating the generation of megalopores of layered structures. The specific BET surface areas are 19 and 10 m^2^/g for Y_2_O_3_ and Y_2_O_3_-NaCl, respectively. Figure 3b displays that the average pore sizes of Y_2_O_3_-NaCl (70 nm) are apparently larger than the non-templated Y_2_O_3_ support (10 nm). These results reveal that the textural properties of Y_2_O_3_ support underwent some transformation after NaCl modification. These results all indicate that we have synthesized two distinctive Y_2_O_3_ supports with similar textural properties.

The bulk concentrations of Pt for as-synthesized Pt/Y_2_O_3_-NaCl (0.68 wt %) and Pt/Y_2_O_3_ (0.62 wt %) were determined by ICP-AES (Table 1), and very close to the designed value (0.7 wt %), verifying the successful deposition of Pt during the synthesis process. The detailed morphology of as-synthesized Pt/Y_2_O_3_-NaCl and Pt/Y_2_O_3_ catalysts was investigated through TEM measurements. The TEM images in Figure 4 show the smaller Pt nanoparticles (about 5.5 nm) in a fresh Pt/Y_2_O_3_-NaCl sample are evenly dispersed on the surface of the NaCl-templated Y_2_O_3_ matrix. The lattice spacing has been measured to confirm Pt (111) of metallic Pt species. However, for Pt/Y_2_O_3_ catalyst, the PtO_2_ (100) and Pt (111) lattice fringes were observed in HRTEM image, indicating that Pt species were mixed phase anchoring on support; the non-uniform nanoparticles dispersion with both small-size nanoparticles and bulky aggregates are intuitively observed (22.9 ± 10.8 nm). Figure 4d–j further confirms the Pt distribution on Y_2_O_3_ supports with and without NaCl oriented via HAADF-STEM EDS mappings. We suppose that the flatter surfaces facilitate the dispersion of Pt, while the irregular surface offered a driving force of Pt gathering. The crystal structures of Y_2_O_3_ supports and Pt/Y_2_O_3_ catalysts were measured and analyzed by XRD. In Figure 4c, the obvious diffraction peaks at 29.2º, 33.8º, 48.5º, and 57.6º can be attributed to the (222), (400), (440), and (622) planes of Y_2_O_3_ (PDF-#-41-1105), respectively. In addition, the Y_2_O_3_-NaCl sample displays sharper reflections than pure Y_2_O_3_ support, implying their better crystallinity with a NaCl template. For fresh Pt/Y_2_O_3_-NaCl and Pt/Y_2_O_3_ samples, no obvious diffraction peak of Pt at 39.8º ascribed to (332) plane (PDF-#-04-0802) was detected or overlapped with the peak of the Y_2_O_3_ supports, and no other characteristic peaks of crystallized Pt species, such as PtO or PtO_2_, were observed on our catalyst (Figure 4h). Combined with the above TEM results, we confirmed the enhanced dispersion of Pt species on NaCl-templated support, and such dispersion difference of the supported Pt species probably results in completely different catalytic activities towards CO oxidation reaction. 

The local structures—including electronic and coordination structures of the supported platinum species—were analyzed with the help of XAFS experiments. The XANES profile of Pt/Y_2_O_3_-NaCl in Figure 5a shows a similar edge shape with PtO_2_ standard, demonstrating its fully oxidized Pt^4+^ state before CO oxidation reaction. However, the profile of the Pt/Y_2_O_3_ sample presents a relatively low valence of 1.8 as a result of the generation of large-size metallic Pt particles on non-templated Y_2_O_3_ support. The EXAFS fitting results on these two fresh samples are shown in Figure 5b and Table 2. A prominent first Pt-O (R ≈ 2.0 Å) shell and a secondary Pt-Pt (R ≈ 2.8 Å) shell with lower intensity were observed for Pt/Y_2_O_3_-NaCl, implying its lower crystallinity. As for Pt/Y_2_O_3_, the first Pt-O (R ≈ 2.0 Å) shell displayed lower intensity, while the secondary Pt-Pt (R ≈ 2.8 Å) shell and further shells displayed higher intensities than Pt/Y_2_O_3_-NaCl, implying the formation of high crystallinity Pt particles. These XAFS findings are in good accordance with the previously mentioned XRD and TEM results (Figure 4). Based on the above characterizations and corresponding analysis—including XRD, TEM, and XAFS measurements towards our newly synthesized catalysts—we have verified that the small-size platinum dioxide species were evenly dispersed on the NaCl-templated Y_2_O_3_ support; while platinum-containing species, such as small-size nanoparticles and some bulky aggregates, were unevenly distributed on the surface of non-templated Y_2_O_3_ support, indicating that the application of NaCl as a template to optimize the support structure can significantly improve the dispersion of the supported Pt species. This may have a positive effect on catalytic activity in the CO oxidation reaction.

### 3.2. Catalytic Performance

CO oxidation reaction was selected as a probe reaction to evaluate the different catalytic behaviors of our Pt/Y_2_O_3_-NaCl and Pt/Y_2_O_3_ catalysts at the same space velocity of 120,000 mL·g_cat_^−1^·h^−1^. The supported Pt catalysts were pretreated at 300 °C for almost 30 min under 5% H_2_/N_2_ atmosphere before testing. As shown in Figure 6a, the initial conversion temperature of Pt/Y_2_O_3_-NaCl (188 °C) was lower than that of Pt/Y_2_O_3_ catalyst (215 °C), and with the complete CO conversions ranged from 120 °C to 190 °C for the corresponding catalysts. As expected, neither the NaCl-templated Y_2_O_3_ support nor the non-templated Y_2_O_3_ support possessed any activity in CO oxidation. Furthermore, the ‘light-off’ temperature of 50% CO conversion for NaCl-templated Y_2_O_3_ supported Pt catalyst reached 185 °C, which was significantly lower than that of non-templated Y_2_O_3_ supported Pt catalyst (211 °C), and even lower than those of the previously reported non-cerium oxide rare earth element oxides supported noble catalysts, such as Pt/La_2_O_3_ (218 °C, 6000 mL·g_cat_^−1^·h^−1^) [21] and Au/Y2O3 (285 °C, 21,000 mL·g_cat_^−1^·h^−1^) [27], revealing the noteworthily catalytic performance of our NaCl modified Pt/Y_2_O_3_ in CO oxidation reaction. For kinetic studies, the apparent activation energies (*E*_a_) of the two catalysts were calculated according to Arrhenius plots at very low CO conversions (<20%) to exclude mass and heat transfer limitations [30]. In Figure 6b, the *E*_a_ value of Pt/Y_2_O_3_-NaCl catalyst (70.6 kJ·mol^−1^) was similar to Pt/Y_2_O_3_ catalyst (76.4 kJ·mol^−1^), implying that the two catalysts probably catalyze via the same reaction mechanism or active site. On the other hand, both Pt/Y_2_O_3_-NaCl or Pt/Y_2_O_3_ exhibited outstanding thermostability at 180 °C or 200 °C for more than 10 h of stability testing with a high space velocity of 120,000 mL·g_cat_^−1^·h^−1^ (Figure 6c,d).

### 3.3. Structural Characterization of Used Platinum-Yttrium Oxide Catalysts

TEM images of used platinum-yttrium oxide catalysts are shown in Figure 7. For Pt/Y_2_O_3_, the nonuniformly aggregated platinum particles were clearly observed, while very small-size platinum nanoparticles with an average size of nearly 7.3 nm were evenly dispersed on the surface of our NaCl-templated Y_2_O_3_ support (Figure 7g) with Pt (111) crystal lattice, and with just a slight growth of such platinum nanoparticles even after CO oxidation reaction, indicating the relatively high stability of platinum nanoparticles supported on nanosheet Y_2_O_3_ support, and further illustrating the importance of NaCl template modification. The particle size of Pt/Y_2_O_3_ catalyst was also increased (29.8 nm) along with Pt (111) and PtO_2_ (110) crystal lattices compared with fresh samples. For the two catalysts, both the particle size and the dispersion changed slightly, which gives a hint that the Y_2_O_3_ supported catalysts were relatively stable during CO oxidation process. Just as we expected, the XRD patterns of the used Pt/Y_2_O_3_-NaCl and Pt/Y_2_O_3_ samples (Figure 7h) still show nothing about any Pt species such as Pt or PtO_2_ while obvious diffraction peaks were observed for Y_2_O_3_ (PDF-#-41-1105) at 29.2º, 33.8º, 48.5º, and 57.6º or an overlapped diffraction peak for the Pt (1, 1, 1) plane at 39.8º. 

In order to interpret the real state of our catalysts before and after the CO oxidation reaction, the XAFS data were collected by using a stainless reactor with two globe valves and tableting the reduced or used samples in a glove box under N_2_ atmosphere. The XANES profiles in Figure 8a of the reduced and used samples display edge jump energy similar to Pt foil, mainly implying the existence of metallic platinum in these samples. Under the help of various standards (Pt foil for Pt^0^, PtO_2_ for Pt^4+^), the oxidation states of platinum species of the two catalysts during the whole reaction process were determined by the linear combination fit. As shown in Figure 7c, the valance of platinum decreased dramatically once reduced under H_2_ atmosphere and then slightly increased during the CO oxidation reaction process for both the two catalysts, while they did not achieve the valence state of fresh samples, manifesting the stability of these catalysts. The related EXAFS fitting results in Figure 8b and Table 3 clearly display the decrease in Pt-O shell (R ≈ 1.98 Å) and the increase in Pt-Pt shell (R ≈ 2.76 Å) after H_2_ pretreating. The CN (9.3) of Pt-Pt shell in the Pt/Y_2_O_3_-H_2_ sample was a little larger than that in Pt/Y_2_O_3_-NaCl-H_2_ sample, indicating the larger Pt NPs size in Y_2_O_3_ supports, which were a similar trend in used samples. Notably, a Pt-O shell appeared at ~2.47 Å in Pt/Y_2_O_3_-NaCl-H_2_, and Pt/Y_2_O_3_-NaCl-used samples differing from conventional distance Pt-O shell (2.00 Å) may act as active oxygen species for CO oxidation as a result of the higher activity of CO oxidation in Y_2_O_3_-NaCl supports. The EXAFS fitting curves were showed in Appendix A). Therefore, on the basis of the above TEM, XRD, and XAFS results, we found that the reduced platinum species were easily formed by hydrogen pretreatment and remained nearly unchanged during the catalytic process over the studied Pt/Y_2_O_3_-NaCl and Pt/Y_2_O_3_ catalysts. Furthermore, the NaCl-templated Y_2_O_3_-nanosheet-supported-platinum nanoparticles exhibited noteworthy stability during the whole process.

It is well known that the surface structure of the catalyst plays a non-negligible role in catalytic activity. We further investigated the oxidation states of platinum species in Pt/Y_2_O_3_-NaCl and Pt/Y_2_O_3_ catalysts before and after CO oxidation by XPS analysis (Figure 9). The results show that the characteristic peaks of Pt^4+^ species were found for the two fresh samples, and the peaks of Pt^2+^ species were found for Pt/Y_2_O_3_-NaCl-fresh (Figure 9); combined with the TEM and XAFS results, we confirm that the generation of small-size PtO_2_ clusters or nanoparticles on Y_2_O_3_ nanosheets results in a remarkable catalytic activity for CO oxidation. Specially, the metallic Pt species at 71.4 eV were detected for fresh Pt/Y_2_O_3_ samples, as a result of formation of bulky PtO2 and Pt aggregates [31,32,33], which was consistent with the previous TEM and XAFS results. For the used catalysts, the peaks of metallic Pt species were observed, and the binding energy for Pt/Y_2_O_3_-NaCl was slightly higher than Pt/Y_2_O_3_. By fitting the XPS peaks (Table 4), we found that most oxidized platinum species were reduced to metallic Pt in Pt/Y_2_O_3_-NaCl catalyst, while less than half of the Pt^4+^ species were reduced to metallic Pt in Pt/Y_2_O_3_ catalyst, indicating that the small-sized PtO_2_ nanoparticles were more easily reduced to small-size Pt species, which would be the active species to facilitate the CO oxidation reaction. In addition, the calculated surface Pt concentration for Pt/Y_2_O_3_-NaCl catalyst was close to the designed one, while it was obviously lower than the designed value for Pt/Y_2_O_3_, implying NaCl-assisted synthesis of yttrium oxide nanosheets was more favorable for the dispersion of Pt species.

Furthermore, to gather further insight into the active species on the surface of our catalysts, in situ DRIFTS experiments were performed with the sequence of ‘CO-N_2_-CO-O_2_’. In Appendix A and Figure 10. As shown in Figure 10, the IR bands at 2074 cm^−1^ and 2080 cm^−1^ were assigned to well-coordinate (WC) facet atoms linearly absorbed CO towards Pt nanoparticles, and 2053 cm^−1^ and 2057 cm^−1^ were assigned to under-coordinate (UC) edge atoms linearly absorbed CO [34,35,36,37,38] towards Pt nanoparticles. It is worth noting that Pt/Y_2_O_3_-NaCl displayed stronger CO adsorption intensity than Pt/Y_2_O_3_, implying the larger amounts of WC Pt sites and UC Pt sites in the uniformly dispersed Pt nanoparticles on Y_2_O_3_ nanosheets [39,40,41,42], which boost CO conversion at a relatively lower temperature. The DRIFTS curves all implied that the large bulk Pt or PtO*_x_* phase in Pt/Y_2_O_3_ catalysts exposed less active Pt sites than Pt species in Pt/Y_2_O_3_-NaCl samples, which gave evidence for higher CO oxidation activity in Pt/Y_2_O_3_-NaCl catalysts.

## 4. Conclusions

In summary, a controllable incipient wetness impregnation approach was carried out to immobilize small-size platinum nanoparticles on sodium chloride templated 2D-yttrium oxide nanosheets. The catalyst exhibited noteworthy catalytic activity towards CO oxidation reaction featuring the ‘light-off’ temperature of 188 °C, significantly lower than the non-templated Y_2_O_3_ supported Pt catalyst (214 °C) at a very high space velocity of 120,000 mL·g_cat_^−1^·h^−1^. The XANES/EXAFS, XPS, and in situ DRIFTS results indicated that uniformly dispersed platinum nanoparticles boost the oxidation of CO. It has been proved through research that Pt species can be well dispersed on the surface of relatively flat 2D Y_2_O_3_, which was grown on the surface of the NaCl template. The application of sodium chloride as a template in the synthesis of yttrium oxide nanosheets plays an essential role in the fabrication of highly efficient platinum nanoparticle catalysts. This work provides a general method for synthesizing large flat 2D materials, and the unique 2D topography of support can effectively adjust the dispersion of catalysts.

## Figures and Tables

**Figure 1 nanomaterials-12-02306-f001:**
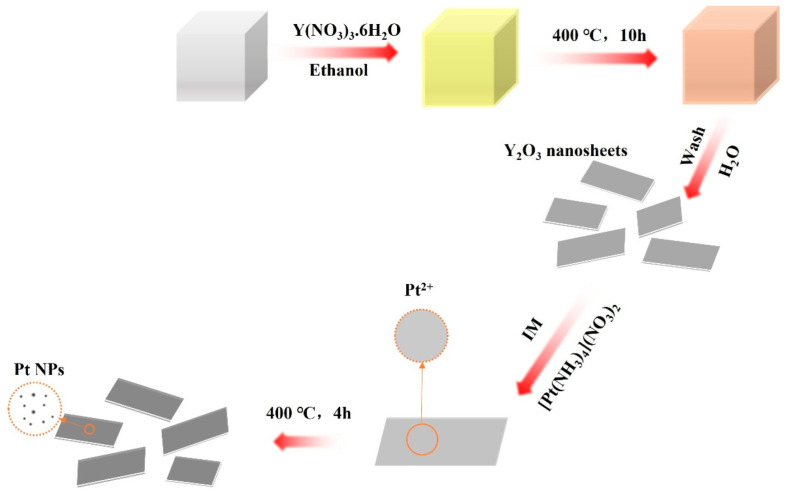
Schematic illustration for preparation of a Pt/Y_2_O_3_-NaCl sample.

**Figure 2 nanomaterials-12-02306-f002:**
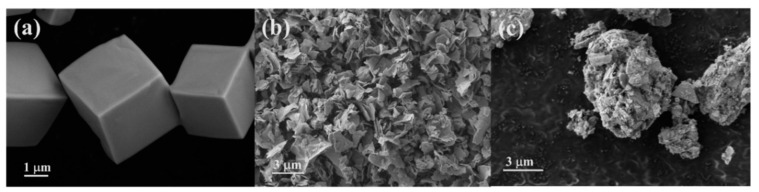
SEM images for (**a**) NaCl, (**b**) Pt/Y_2_O_3_-NaCl, and (**c**) Pt/Y_2_O_3_.

**Figure 3 nanomaterials-12-02306-f003:**
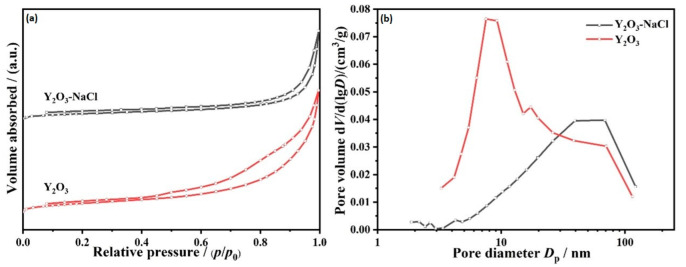
Nitrogen adsorption–desorption isotherms (**a**) and pore size distributions (**b**) of as-prepared Y_2_O_3_-NaCl and Y_2_O_3_ supports.

**Figure 4 nanomaterials-12-02306-f004:**
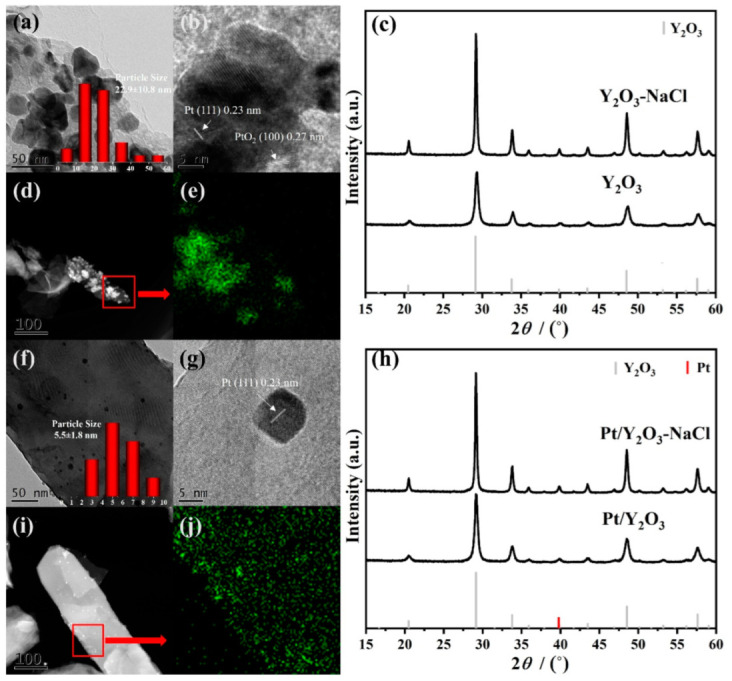
Structural characterization of supports and fresh catalysts. TEM (**a**,**f**) and HRTEM images (**b**,**g**), and HAADF-STEM images (**d**,**i**) and corresponding EDS mapping results (**e**,**j**) of fresh Pt/Y_2_O_3_ (**a**,**b**,**d**,**e**) and Pt/Y_2_O_3_-NaCl (**f**,**g**,**i**,**j**), and XRD patterns of Y_2_O_3_-NaCl and Y_2_O_3_ supports (**c**), and fresh Pt/Y2O3-NaCl and Pt/Y_2_O_3_ catalysts (**h**).

**Figure 5 nanomaterials-12-02306-f005:**
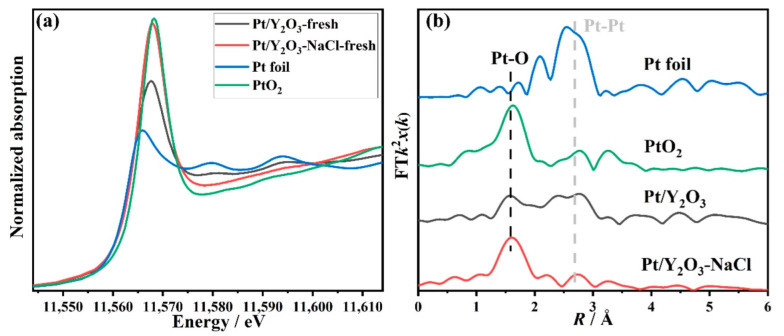
Pt L3-edge XANES profiles (**a**) and EXAFS fitting results (**b**) in R space for fresh Pt/Y_2_O_3_-NaCl and Pt/Y_2_O_3_ catalysts.

**Figure 6 nanomaterials-12-02306-f006:**
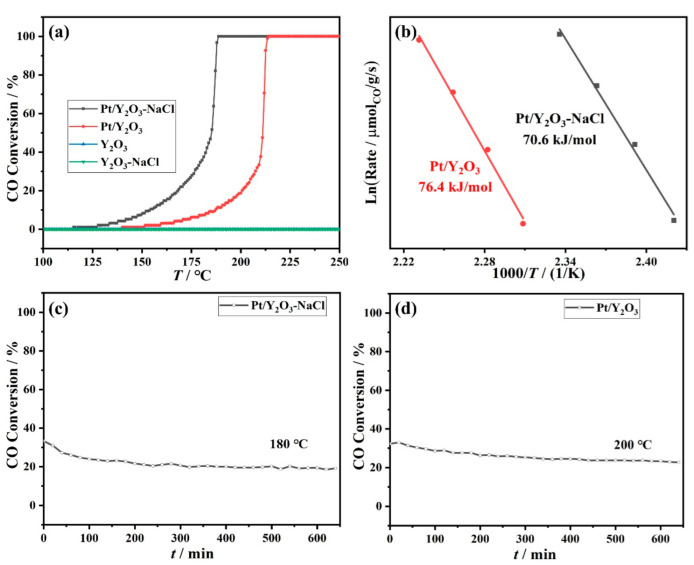
Catalytic performance of Pt/Y_2_O_3_-NaCl and Pt/Y_2_O_3_ catalysts: (**a**) ‘light off’, (**b**) apparent activation energy (*E*_a_), and stability test of (**c**) Pt/Y_2_O_3_-NaCl and (**d**) Pt/Y_2_O_3_ for CO oxidation.

**Figure 7 nanomaterials-12-02306-f007:**
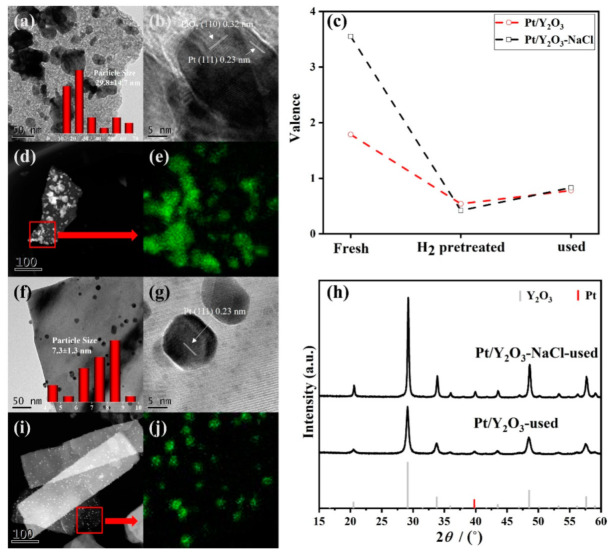
Structural characterization of used catalysts. TEM (**a**,**f**) and HRTEM images (**b**,**g**), and HAADF-STEM images (**d**,**i**) and corresponding EDS mapping results (**e**,**j**) of used Pt/Y_2_O_3_ (**a**,**b**,**d**,**e**) and Pt/Y_2_O_3_-NaCl (**f**,**g**,**i**,**j**), chemical valence evolution obtained from the linear combination fits on XANES profiles (**c**) and XRD patterns (**h**) of Pt/Y_2_O_3_-NaCl and Pt/Y_2_O_3_ catalysts.

**Figure 8 nanomaterials-12-02306-f008:**
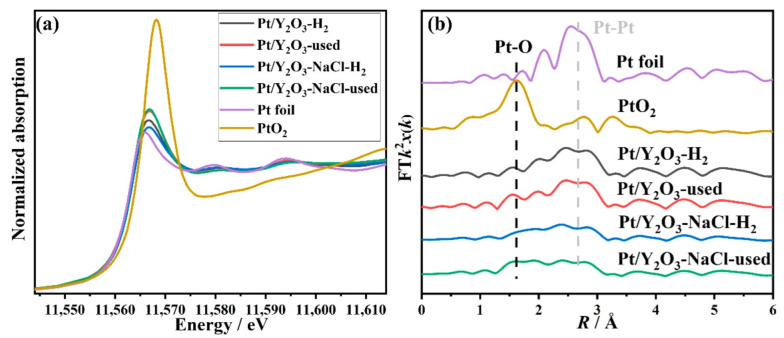
Pt L3-edge XANES profiles (**a**) and EXAFS fitting results (**b**) in R space for reduced and used Pt/Y_2_O_3_-NaCl and Pt/Y_2_O_3_ catalysts.

**Figure 9 nanomaterials-12-02306-f009:**
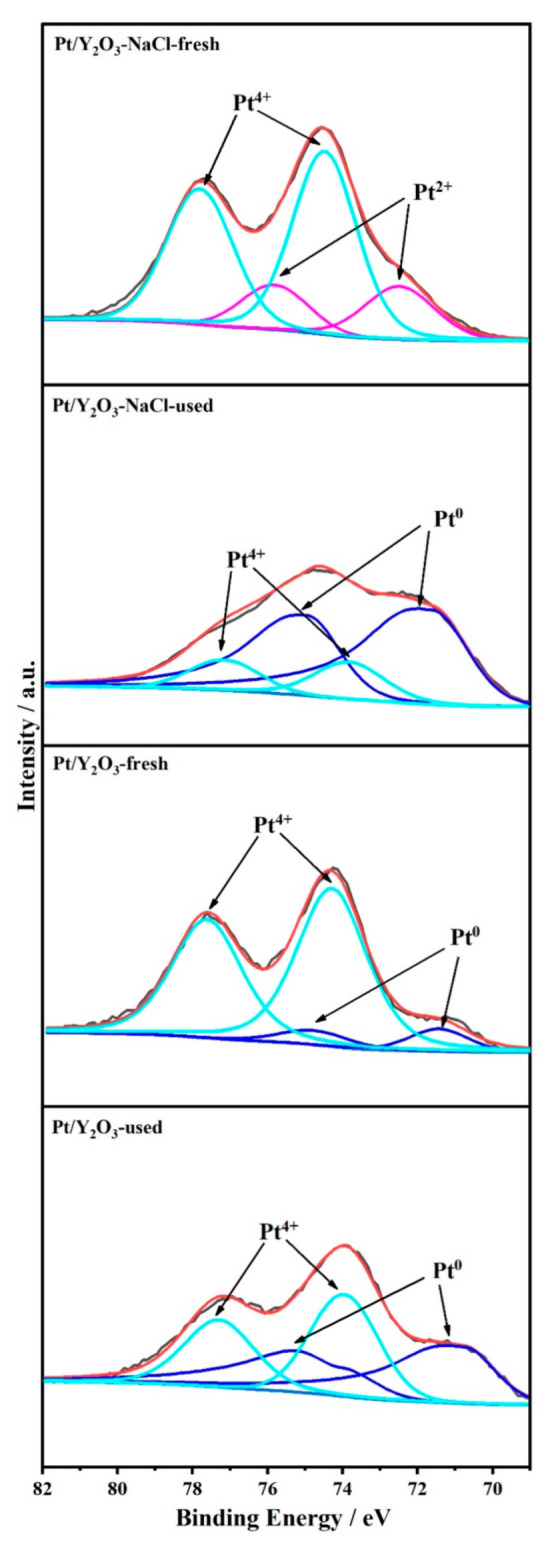
XPS profiles for fresh and used Pt/Y_2_O_3_-NaCl and Pt/Y_2_O_3_ catalysts.

**Figure 10 nanomaterials-12-02306-f010:**
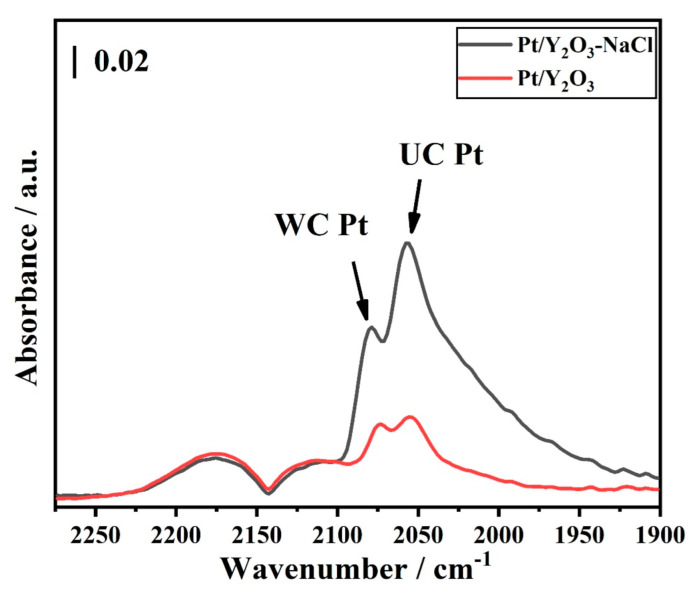
In situ DRIFTS of Pt/Y_2_O_3_ and Pt/Y_2_O_3_-NaCl catalysts after 30 min CO adsorption.

**Table 1 nanomaterials-12-02306-t001:** Metal mass ratios of Pt/Y_2_O_3_ (*C*_Pt*/*Y_), BET specific surface areas (*S*_BET_), BJH pore volume (*V*_p_), lattice constants (*a*) of Y_2_O_3_ and average particle sizes calculated from XRD (*D*_XRD_).

Sample	*C* _Pt/Y_	*S* _BET_	*V* _P_	*a*	*D* _XRD_
	(wt %) ^a^	(m^2^g^−1^) ^b^	(cm^3^g^−1^) ^b^	(Å) ^c^	(nm) ^c^
Y_2_O_3_	-	19	0.07	10.5659(6)	17
Y_2_O_3_-NaCl	-	10	0.05	10.5915(6)	30
Pt/Y_2_O_3_	0.62	-	-	10.6049(6)	18
Pt/Y_2_O_3_-NaCl	0.68	-	-	10.6038(6)	31
Pt/Y_2_O_3_-used	-	-	-	10.6069(6)	19
Pt/Y_2_O_3_-NaCl-used	-	-	-	10.5900(6)	30

^a^ Determined by ICP-AES; ^b^ Calculated from nitrogen adsorption–desorption results; ^c^ Calculated from XRD patterns.

**Table 2 nanomaterials-12-02306-t002:** EXAFS fitting results for fresh Pt/Y_2_O_3_-NaCl and Pt/Y_2_O_3_ catalysts.

Sample	Shell	*CN*	*R* (Å)	*σ*^2^ (Å^2^)	Δ*E*_0_
Pt/Y_2_O_3_-fresh	Pt-O	2.0 ± 0.2	2.00 ± 0.01	0.0024	11.5 ± 1.4
	Pt-Pt	7.0 ± 0.6	2.77 ± 0.01	0.0040	
	Pt-O-O	3.7 ± 1.1	2.81 ± 0.03	0.0076	
	Pt-O-Pt	5.4 ± 1.8	3.24 ± 0.03	0.0094	
Pt/Y_2_O_3_-NaCl-fresh	Pt-O	4.9 ± 0.2	2.01 ± 0.01	0.0039	11.6 ± 1.0
	Pt-Pt	3.2 ± 0.7	2.80 ± 0.01	0.0060	
	Pt-O-O	3.3 ± 1.0	2.99 ± 0.03	0.0094	

*R*: distance; *CN*: coordination number; *σ*^2^: Debye–Waller factor; Δ*E*_0_: inner potential correction.

**Table 3 nanomaterials-12-02306-t003:** EXAFS fitting results for reduced and used Pt/Y_2_O_3_-NaCl and Pt/Y_2_O_3_ catalysts.

Sample	Shell	*CN*	*R* (Å)	*σ*^2^ (Å^2^)	Δ*E*_0_
Pt/Y_2_O_3_-H_2_	Pt-O	1.0 ± 0.2	1.98 ± 0.01	0.0086	8.6 ± 0.9
	Pt-Pt	9.3 ± 0.3	2.76 ± 0.01	0.0053	
	Pt-O-Pt	2.2 ± 1.4	3.09 ± 0.06	0.0157	
Pt/Y_2_O_3_-used	Pt-O	1.1 ± 0.2	1.99 ± 0.01	0.0026	8.1 ± 1.0
	Pt-Pt	7.8 ± 0.3	2.76 ± 0.01	0.0044	
Pt/Y_2_O_3_-NaCl-H_2_	Pt-O	0.6 ± 0.2	2.01 ± 0.02	0.0016	5.7 ± 1.9
	Pt-O1	0.5 ± 0.4	2.46 ± 0.05	0.0034	
	Pt-Pt	7.1 ± 0.4	2.74 ± 0.01	0.0077	
Pt/Y_2_O_3_-NaCl-used	Pt-O	1.4 ± 0.2	1.98 ± 0.01	0.0038	3.1 ± 1.7
	Pt-O1	1.7 ± 0.5	2.47 ± 0.02	0.0047	
	Pt-Pt	6.0 ± 0.3	2.73 ± 0.01	0.0065	

*R*: distance; *CN*: coordination number; *σ*^2^: Debye–Waller factor; Δ*E*_0_: inner potential correction.

**Table 4 nanomaterials-12-02306-t004:** XPS fitting results for fresh and used Pt/Y_2_O_3_-NaCl and Pt/Y_2_O_3_ catalysts.

Sample	*C* _Pt_	*C* _Pt surface_	Pt^0^	Pt^2+^	Pt^4+^
	(at. %) ^a^	(at. %) ^b^	(%) ^b^	(%) ^b^	(%) ^b^
Pt/Y_2_O_3_-fresh	0.62	0.36	9.9	-	90.1
Pt/Y_2_O_3_-NaCl-fresh	0.68	1.22	-	23.9	76.1
Pt/Y_2_O_3_-used	-	0.36	50.0	-	50.0
Pt/Y_2_O_3_-NaCl-used	-	0.94	80.2	-	19.8

^a^: Calculated by ICP; ^b^: fitted by XPS.

## Data Availability

Not applicable.

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
