# Peer review of "NaCl-Templated Ultrathin 2D-Yttria Nanosheets Supported Pt Nanoparticles for Enhancing CO Oxidation Reaction"

_nanomaterials, 2022, doi:10.3390/nano12132306_

Round 1

Reviewer 1 Report

Comments:

The authors present a nice study of the formation of Pt/Y2O3 catalysts, but the study would be improved with more information as well as more discussion of both the synthesis and the results. My detailed comments are below.

1) Can the authors comment more on the nanoparticles of Pt that are formed? From the impregnation method how the particles are formed and are the particles observed Pt or PtO2? This should be indicated in Figure 1.

2) Is it possible to show HRTEM images of the Pt nanoparticles to give some idea of the crystallinity and structure as the XRD does not give this information?

3) Can the authors provide the Scherrer equation analysis for each major diffraction of the Y2O3 for both samples? I would expect some degree of variation in the values of the plates due to their structure, and it will also give a better idea of the difference in the particle sizes.

4) Can the authors comment on the different size distributions for the Pt on the templated Y2O3 and the non-template ones? Why would this occur? What is driving the particle formation and thus the growth/aggregation without the template?

5) Can the authors provide the experimental characterization of the Pt catalysts that were pre-treated at 300C under 5% H2/N2? How much different are these from the as synthesized materials?

6) On page 10 line 312 to 313 the authors mention or overlapped diffraction peak for Pt (111), I presume this is meant to say that Pt(111) is still not observed. What about PtO2? Or is it observed?

7) The authors indicate that the used samples are now Pt, are they reduced to Pt in the pre-treatment? This would be good to establish as the particles at this point might look more like the used samples.

8) The authors should report particle size distributions for all the samples to make it easier to compare between the samples.

9) In the author list is has an ‘and’ after Lina Li, please remove that.

Reviewer 2 Report

The manuscript is interesting and it requires some moderate revisions:

1)English grammar should be double-checked;

2)in general, the results discussion must be enlarged and more comparisons with literature data should be reported. The authors should go more in depth with the interpretation of the results;

3)the advancement of knowledge reached by the findings of the study must be underlined in the conclusion section;

4)the unit of measure of each parameter should follow the parameter itself when it appears in the text;

5)the practical implication of this study are not well covered by the authors.

Round 2

Reviewer 1 Report

I am happy with the changes. One minor point. On Page 5 the authors mention the average size, they should mention this is the average crystallite size taken from XRD.
